# Factors associated with differential seropositivity to *Leptospira interrogans* and *Leptospira kirschneri* in a high transmission urban setting for leptospirosis in Brazil

**Daiana de Oliveira**[1,2], **Hussein Khalil**[3], **Fabiana Almerinda G. Palma**[1], **Roberta Santana**[1], **Nivison Nery, Jr**[1,2], **Juan C. Quintero-Vélez**[4,5,6], **Caio Graco Zeppelini**[7], **Gielson Almeida do Sacramento**[2], **Jaqueline S. Cruz**[2], **Ricardo Lustosa**[1], **Igor Santana Ferreira**[1], **Ticiana Carvalho-Pereira**[1], **Peter J. Diggle**[8], **Elsio A. Wunder, Jr**[2,9,10], **Albert I. Ko**[2,9], **Yeimi Alzate Lopez**[1], **Mike Begon**[11], **Mitermayer G. Reis**[2,9,12], **Federico Costa**[1,2,9]*

1 Instituto da Saúde Coletiva, Universidade Federal da Bahia, Salvador, Bahia, Brasil, 2 Instituto Gonçalo Moniz, Fundação Oswaldo Cruz, Ministério da Saúde, Salvador, Bahia, Brasil, 3 Department of Wildlife, Fish, and Environmental Studies, Swedish University of Agricultural Sciences, Umeå, Sweden, 4 Grupo de Ciencias Veterinarias—Centauro, Facultad de Ciencias Agrarias, Universidad de Antioquia, Medellín, Colombia, 5 Grupo de Investigación Microbiología Básica y Aplicada, Universidad de Antioquia, Medellín, Colombia, 6 Grupo de Epidemiología, Universidad de Antioquia, Medellín, Colombia, 7 Programa de Pós-Graduação em Ecologia: Teoria, Aplicações e Valores, Universidade Federal da Bahia, Salvador, Brazil, 8 Centre for Health Informatics, Computing, and Statistics, Lancaster University, Lancaster, United Kingdom, 9 Department of Epidemiology of Microbial Diseases, Yale School of Public Health, New Haven, Connecticut, United States of America, 10 Department of Pathobiology and Veterinary Science, University of Connecticut, Storrs, Connecticut, United States of America, 11 Institute of Integrative Biology, University of Liverpool, Liverpool, United Kingdom, 12 Faculdade de Medicina, Universidade Federal da Bahia, Salvador, Bahia, Brasil

* federico.costa@ufba.br

## Abstract

### Background

Leptospirosis is a zoonosis caused by pathogenic species of bacteria belonging to the genus *Leptospira*. Most studies infer the epidemiological patterns of a single serogroup or aggregate all serogroups to estimate overall seropositivity, thus not exploring the risks of exposure to distinct serogroups. The present study aims to delineate the demographic, socioeconomic and environmental factors associated with seropositivity of *Leptospira* serogroup Icterohaemorraghiae and serogroup Cynopteri in an urban high transmission setting for leptospirosis in Brazil.

### Methods/Principal findings

We performed a cross-sectional serological study in five informal urban communities in the city of Salvador, Brazil. During the years 2018, 2020 2021, we recruited 2.808 residents and collected blood samples for serological analysis using microagglutination assays. We used a fixed-effect multinomial logistic regression model to identify risk factors associated with seropositivity for each serogroup. Seropositivity to Cynopteri increased with each year of age (OR 1.03; 95% CI 1.01–1.06) and was higher in those living in houses with unplastered

**Data Availability Statement:** All relevant data are in the manuscript and its supporting information files.

**Funding:** The study was funded by Medical Research Council (UK) Grant number: MR/P024084/1 to MB; Fundação de Amparo à Pesquisa do Estado da Bahia (BR) Grant number: 10206/2015, Wellcome Trust (UK) Grant number: 102330/Z/13/Z to FC; and National Institutes of Health (US) Grant numbers: R01 TW009504 and R01 AI121207 to AIK. The funders had no role in study design, data collection and analysis, decision to publish, or preparation of the manuscript.

**Competing interests:** The authors have declared that no competing interests exist.

walls (exposed brick) (OR 1.68; 95% CI 1.09–2.59) and where cats were present near the household (OR 2.00; 95% CI 1.03–3.88). Seropositivity to Icterohaemorrhagiae also increased with each year of age (OR 1.02; 95% CI 1.01–1.03) and was higher in males (OR 1.51; 95% CI 1.09–2.10), in those with work-related exposures (OR 1.71; 95% CI 1.10–2.66) or who had contact with sewage (OR 1.42; 95% CI 1.00–2.03). Spatial analysis showed differences in distribution of seropositivity to serogroups Icterohaemorrhagiae and Cynopteri within the five districts where study communities were situated.

## Conclusions/Significance

Our data suggest distinct epidemiological patterns associated with the Icterohaemorrhagiae and Cynopteri serogroups in the urban environment at high risk for leptospirosis and with differences in spatial niches. We emphasize the need for studies that accurately identify the different pathogenic serogroups that circulate and infect residents of low-income areas.

### Author summary

In this study, we investigated specific epidemiological patterns related to the Icterohaemorrhagiae and Cynopteri serogroups in urban environments at high risk for leptospirosis. Our results indicate that seropositivity for the Cynopteri serogroup increases with age. Seropositivity is also higher in places where houses have unplastered walls and where there are cats in the vicinity of the residences. Seropositivity for the Icterohaemorrhagiae serogroup also increases with age, and is more prevalent in men and in individuals exposed to work or who have contact with sewage. Spatial analysis reveals variations in the distribution of seropositivity for these serogroups within the five urban areas studied. We emphasize the importance of evaluating the epidemiological pattern for the different *Leptospira* serogroups, since each serogroup may be associated with different animal reservoirs. Our data suggest that the Icterohaemorrhagiae and Cynopteri serogroups present different epidemiological pattern, thus highlighting the need to understand the complex interactions between serogroups and reservoirs. These findings not only contribute to our understanding of the epidemiology of leptospirosis, but also have practical public health implications by helping to identify risk factors and thereby develop more effective preventive measures.

## Introduction

Leptospirosis is a potentially severe infectious disease that affects animals (both domestic and wild) and humans [1]. The disease is caused by pathogenic species of the genus *Leptospira* and results in a wide array of clinical manifestations in humans [2]. Symptoms can range from asymptomatic infection and light fever to severe manifestations with risk of death [3]. Over a million cases occur globally every year, causing over sixty thousand deaths [4]. The majority of the disease burden occurs in developing nations, in local environments that lack adequate infrastructure and health services [5]. Human infection occurs through direct contact with urine or contaminated secretions from infected animals, or indirectly through contact with contaminated soil and water [1,6].

The genus *Leptospira* comprises 72 species and over 300 serovars, classified in more than 20 serogroups [7–10]. The microscopic agglutination test (MAT) is the reference serological assay for the diagnosis of leptospirosis. Despite its limitations, the MAT can provide the presumptive identity of the infecting serogroup, contributing for spatiotemporal epidemiological studies and reservoir identification when isolation of local strains is not available [11]. However, isolation or specific laboratory techniques like DNA sequence analysis are needed for definitive serotyping [12,13]. Some serovars can infect multiple animal species while others are more adapted to specific animal hosts [2]. This host preference might explain why some serovars/serogroups are more frequent in certain geographic areas. The diversity of reservoirs, serovars and serogroups, together with environmental and socio-behavioral determinants, poses a challenge to the characterization of the epidemiologic patterns of leptospirosis. These factors can influence the risk of transmission and consequently increase incidence and cause outbreaks [14].

Historically, tropical and subtropical nations have experienced seasonal leptospirosis incidence peaks associated with heavy rainfall and flooding events during the rainy season [14–17]. The rapid, informal expansion of large urban centers, coupled with inadequate housing and sanitation, has resulted in environmental and social conditions that are highly favorable to disease hosts and pathogen transmission [18]. Over 33% of the world's population, and around 28% if Brazil's population live in urban poor communities [19] where, because of the deep social inequalities, there is a direct correlation between vulnerable environments, disaster risk and public health problems [20,21]. Historically, in the city of Salvador, where most of the population is of black and mixed race, people with lower income live in the peripheral areas of cities where access to basic rights is often absent and the prevalence of health problems such as leptospirosis is greater than in other areas of the city.

Studies performed in the past two decades in Salvador, Brazil, have reported *Leptospira interrogans* serogroup Icterohaemorrhagiae serovar Copenhageni (strain Fiocruz L1-130) as the serogroup responsible for over 95% of the clinical cases recorded throughout the city [14], as well as asymptomatic infections in residents of the poor urban community of Pau da Lima within Salvador [22,23]. Previous studies in the same community have shown that the animal reservoir of the serovar Copenhageni and serogroup Icterohaemorrhagiae is the brown rat (*Rattus norvegicus*) [24,25]. These studies identified the seroincidence of *Leptospira kirschneri*, serogroup Cynopteri (strain 3522C) as the second highest in residents of urban communities [26]. However, the animal reservoir of these serogroups and the determinants of its transmission remain unknown.

Investigating the *Leptospira* serologic profile of the residents in urban communities, accounting for serogroup-specific risk factors, will contribute to a better understanding of the risk determinants of urban infection and inform the development of more specific control measures. The present study aims to identify demographic, socioeconomic, and environmental factors associated with seropositivity of *Leptospira interrogans* Icterohaemorrhagiae (strain Fiocruz L1-130) and *Leptospira kirschneri* Cynopteri (strain 3522C) in residents of five urban low-income communities in Salvador, Brazil.

## Methods

### Ethics statement

The study was approved by the Ethics Committee of Fundação Oswaldo Cruz, Instituto Gonçalo Moniz (CAAE 45217415.4.0000.0040), the Ethics Committee for Research of Instituto de Saúde Coletiva (UFBA, 041 / 17–2.245.914.17–2.245.914) and the Institutional Review Board of Yale University (no. 2000031554). Those willing to participate signed an informed consent

form and had a small blood sample drawn by a trained phlebotomist. Underage participants were enrolled with informed consent written of their legal guardian.

## Study design

We performed a cross-sectional study in five urban poor communities: Pau da Lima (PL), Alto do Cabrito (AdC), Marechal Rondon (MR), Nova Constituinte (NC) and Rio Sena (RS) in the city of Salvador, Bahia, Brazil (Fig 1). In PL, we collected samples during the period from November 2020 to February 2021. Previous studies in the area indicated an annual leptospiral incidence of 37.8 infections per thousand [22]. In the four other communities, namely: AdC, MR, NC and RS, sampling occurred from April to September 2018. All five communities are considered low-income, with precarious sanitation and infrastructure.

Eligible individuals were those ≥5 years of age, who slept 3 nights per week in a household. We applied a structured questionnaire to collect information on individual characteristics of the participant, domestic and peri-domestic environmental characteristics, exposure to sources of environmental contamination, and the presence of potential animal reservoirs.

Serological analysis was performed using the Microscopic Agglutination Test (MAT), the reference assay for diagnosis of human leptospirosis, as previously described [18]. MAT is based on dark-field microscopy detection of serum agglutination samples from an individual (antibodies) with live *Leptospira* antibodies. Samples were tested against a panel of seven antigens, including five reference strains (WHO Collaborative Laboratory for Leptospirosis, Royal Tropical Institute, Holland): *L. kirschneri* serogroups Cynopteri serovar Cynopteri strain 3522C and Grippothyphosa serovar Grippothyphosa strain Duyster; *L. interrogans* serogroups

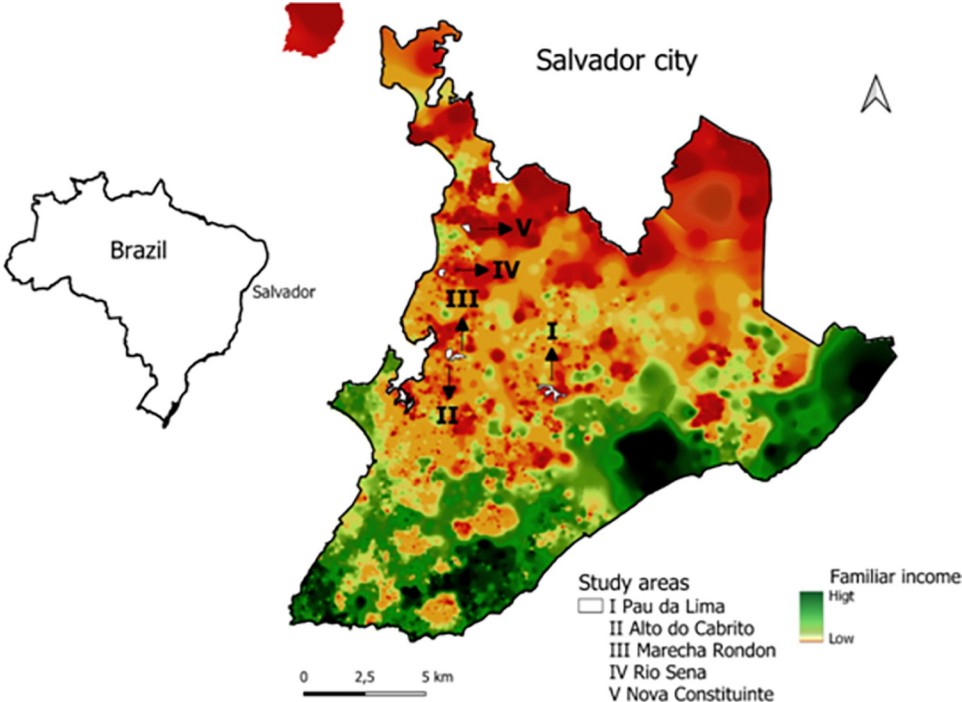

**Fig 1.** Urban communities in the city of Salvador, Brazil. Location of study sites (white) within the city. Distribution of family income, high (green) and low (orange). Shapefiles and data used for the base map were obtained from open and publicly accessible base of IBGE—Instituto Brasileiro de Geografia e Estatística [Brazilian Institute of Geography and Statistics]. Direct link to the base layer of the map: https://www.ibge.gov.br/geociencias/downloads-geociencias.html.

Canicola serovar Canicola strain H. Utrecht IV and Autumnalis serovar Autumnalis strain Akiyami A; and *L. borgpetersenii* serogroup Ballum serovar Ballum strain MUS 127. The panel also included two local clinical isolates: *L. interrogans* serogroup Icterohaemorrhagiae serovar Copenhageni strain Fiocruz L1-130 and *L. santarosai* serogroup Shermani strain LV3954 [22]. We performed association analyses only for serogroups Icterohaemorrhagiae strain Fiocruz L1-130 and Cynopteri strain 3522C, which were the most prevalent strains in our analyses, and presented adequate sample sizes for estimating associations.

## Outcomes, exposures, and covariates

The outcome of positivity for leptospirosis was defined as seropositivity against serogroups Cynopteri or Icterohaemorrhagiae. The interpretation of MAT results for serogroup classification is limited due to the possibility of cross-reactivity between different serovars, lack of representative strains for the local circulating serogroup, and subjectivity of the interpretation of the agglutination [27]. A result was considered positive if 50% or more leptospires were agglutinated by MAT with a titer of $\geq$1:50. When agglutination was observed at a dilution of 1:100, the sample was titrated in serial two-fold dilutions to determine the highest positive titer [26]. In this study, the presumptive definition of the serogroup was established based on the serogroup that exhibited an antibody titer at least one dilution higher than any other serogroup. In cases where serogroups showed equal titers, the presumptive serogroup was classified as mixed and not considered in the final analysis. Additionally, samples that did not show significant reactivity, with an antibody titer below 1:50 in the tested serogroups, were considered negative.

The main explanatory variables at the individual-level were: age in years (as a quantitative variable); gender; ethnicity; highest education level achieved; receipt of the Federal government stimulus program (Bolsa Família); employment status; work involving contact with sewer/garbage/construction materials; use of protective boots; contact with mud in the last 12 months; walking barefoot outside; and cleaning sewer canals in the last 12 months. Additional environmental explanatory variables included: proximity of the household to open sewers; paved access to the household; household with a backyard; unplastered walls (a proxy for quality of construction); garbage accumulation near the household; and presence of domestic animals (cats, dogs, and chickens).

## Spatial descriptive analysis

QGIS version 2.18.20 was used to construct a georeferenced database from satellite image WorldView-3 May 2017, at 31 cm resolution. The study team identified households within the study site and marked their positions onto hard copy 1:1,500 scale maps. The image World-View-3 was acquired by the research project / Institute Gonçalo de Moniz—IGM—Fiocruz Bahia from the company Stamap, with disclosure permitted referencing the Copyrights of DigitalGlobe images. We used bivariate Kernel Density Estimation (KDE) [28], to evaluate possible territorial segregation of serogroups Icterohaemorrhagiae and Cynopteri. The KDE is defined as fhat(x) = $n^{-1}h^{-2} \sum k\{(x-x_i)/h\}$, where k(u) is the kernel function (a bivariate probability density), x denotes any location within the study-area, $x_1,\ldots,x_n$ are the data-locations and h is the bandwidth. The KDE delivers smoothed spatial distributions of the density of human cases of serogroups Icterohaemorrhagiae and Cynopteri, we set the bandwidth at 30 meters, based on the average distance typically traveled by rats per day [29], to give a possible exposure factor of households to each serogroup. To determine the smoothed, population-adjusted risk distribution for each of the serogroups Icterohaemorrhagiae and Cynopteri we then calculated the ratio between the KDE for households with positive cases and the KDE for all households

evaluated. Boundary polygons and the data for the production of the Salvador income map were downloaded from the publicly accessible data base of IBGE—Instituto Brasileiro de Geografia e Estatística [Brazilian Institute of Geography and Statistics], using the Geosciences platform that can be accessed at https://www.ibge.gov.br/geociencias/downloads-geociencias. html. The maps were created by the authors using QGIS 2.18 software.

### Statistical analysis

We performed a descriptive analysis using relative and absolute frequencies for qualitative variables and median and interquartile range for quantitative variables. A fixed-effect multinomial logistic regression model was used to estimate potential factors associated with seropositivity for each of the two serogroups, Icterohaemorrhagiae and Cynopteri. Variables considered in the multivariable analysis were those with $p < 0.15$ in single-factor analyses. The multivariable analysis used a stepwise method based on the purposeful selection of variables according to both statistical and research criteria (biological plausibility) [22,23]. Also, "area" was included in the final model as a five-level categorical fixed effect. The final model, constrained by the inclusion of variables selected on the grounds of biological plausibility was chosen to minimize Akaike's Information Criteria (AIC). The linearity assumption was confirmed by visual inspection before the inclusion of each quantitative variables in bivariate and multivariate models. We then checked for high levels of correlation among the selected variables in the final model using a Variance Inflation Factor (VIF), but all VIF values were <5 and thus no previously selected variables were excluded. All statistical procedures were performed in R studio, using tidyverse, lme4 and nmet packages [30–33]

### Results

The study included 2,808 residents of five neighborhoods located in Salvador, Brazil (Fig 1); of which 1,512 were from Pau da Lima, 332 from Marechal Rondon, 367 from Alto do Cabrito, 304 from Nova Constituinte and 293 from Rio Sena (Table 1). The population studied was predominantly female (58%; 1,620/2,808) and self-declared black (51%; 1,422/2,808) (Table 1).

The overall unadjusted seroprevalence was 9% (253/2,808, 95% CI 8.0% - 10.1%). Of the 253 infected, 18% (45/253) were seropositive to serogroup Cynopteri, of which 43 (96%) reacted to titers of 50 to 400, while the remaining 2 (4%) reacted to titers of 800 to 6400. Also, 82% (208/253) were seropositive to serogroup Icterohaemorrhagiae, of which 190 (91%) reacted to titers of 50 to 400, while the remaining 18 (9%) reacted to titers of 800 to 1600. The seropositivity for the other tested serogroups (Canicola, Shermani, Autumnalis, Ballum, Grippotyphosa) did not exceed 2%, while the mixed result was 7% (S1 and S2 Tables). Seropositivity to serogroup Icterohaemorrhagiae was lowest in AdC (6.5%) and highest in RS (7.8%). Seroprevalence of Cynopteri ranged between 1.0% in NC and 2.1% in MR (Table 1). Seroprevalence of Icterohaemorrhagiae increased with age (OR 1.03 per year, 95% CI 1.02–1.04) as shown in S3 Table and described by age group in Fig 2. Females were less likely to be infected than males (OR 0.69, 95% CI 0.52–0.91, S3 Table). This pattern was not observed for serogroup Cynopteri (OR 1.07, 95% CI 0.59–1.99), which presented higher seroprevalence in the age group >45 years (Fig 2).

The final multivariate multinomial analysis (Table 2), which includes data for residents > 18 years of age to investigate work-related exposures (n = 2,009), identified the following variables as significantly associated with seropositivity against Cynopteri: age in years (as a quantitative variable, OR 1.03, 95% CI 1.01–1.06), lives in house with unplastered walls (exposed brick, OR 1.68, 95% CI 1.09–2.59) and presence of cats near the household (OR 2.00, 95% CI 1.03–3.88). In addition, gender (males>females, OR 1.51, 95% CI 1.09–2.10), age in

**Table 1. Descriptive statistics for demography and socioenvironmental variables by area.**

| Characteristic | N | Total N = 2,808[1] | Alto do Cabrito N = 367[1] | Marechal Rondon N = 332[1] | Nova Constituinte N = 304[1] | Pau da Lima N = 1,512[1] | Rio Sena N = 293[1] | p-value[2] |
|---|---|---|---|---|---|---|---|---|
| **Sociodemographic** | | | | | | | | |
| Age (years) | 2,81 | | | | | | | <**0.001** |
| 0 to 16 | | 666 (24) | 59 (16) | 48 (14) | 72 (24) | 395 (26) | 92 (31) | |
| 17 to 29 | | 708 (25) | 102 (28) | 74 (22) | 74 (24) | 386 (26) | 72 (25) | |
| 30 to 44 | | 721 (26) | 97 (26) | 84 (25) | 87 (29) | 388 (26) | 65 (22) | |
| > 45 | | 713 (25) | 109 (30) | 126 (38) | 71 (23) | 343 (23) | 64 (22) | |
| Sex | 2,81 | | | | | | | 0.94 |
| Male | | 1,188 (42) | 159 (43) | 134 (40) | 130 (43) | 639 (42) | 126 (43) | |
| Female | | 1,620 (58) | 208 (57) | 198 (60) | 174 (57) | 873 (58) | 167 (57) | |
| Ethnicity | 2,77 | | | | | | | <**0.001** |
| Brown | | 1,171 (42) | 146 (41) | 148 (46) | 115 (38) | 635 (42) | 127 (45) | |
| Black | | 1,422 (51) | 191 (54) | 151 (47) | 169 (57) | 781 (52) | 130 (46) | |
| White | | 155 (5.6) | 16 (4.5) | 21 (6.6) | 15 (5.0) | 76 (5.0) | 27 (9.5) | |
| Others | | 20 (0.7) | 0 (0) | 0 (0) | 0 (0) | 20 (1.3) | 0 (0) | |
| Education | 2,81 | | | | | | | <**0.001** |
| 5 or less | | 769 (27) | 87 (24) | 84 (25) | 80 (26) | 430 (28) | 88 (30) | |
| 5 to 9 | | 919 (33) | 94 (26) | 103 (31) | 87 (29) | 515 (34) | 120 (41) | |
| 9 to 12 | | 945 (34) | 159 (43) | 121 (36) | 116 (38) | 479 (32) | 70 (24) | |
| Higher Education | | 61 (2.2) | 17 (4.6) | 7 (2.1) | 10 (3.3) | 25 (1.7) | 2 (0.7) | |
| Never studied | | 114 (4.1) | 10 (2.7) | 17 (5.1) | 11 (3.6) | 63 (4.2) | 13 (4.4) | |
| Employment | 2,81 | 982 (35) | 127 (35) | 105 (32) | 98 (32) | 568 (38) | 84 (29) | **0.016** |
| **Risk Actions** | | | | | | | | |
| Risk occupation | 2,81 | 215 (7.7) | 24 (6.5) | 37 (11) | 15 (4.9) | 121 (8.0) | 18 (6.1) | **0.029** |
| Walk barefoot | 2,81 | 1,046 (37) | 141 (38) | 105 (32) | 101 (33) | 561 (37) | 138 (47) | <**0.001** |
| Use of boots | 2,81 | 559 (20) | 60 (16) | 59 (18) | 58 (19) | 341 (23) | 41 (14) | **0.002** |
| Cleaned sewage | 2,49 | 253 (10) | 27 (8.2) | 43 (14) | 40 (15) | 113 (8.5) | 30 (12) | <**0.001** |
| Sewage contact | 2,81 | 600 (21) | 48 (13) | 97 (29) | 56 (18) | 332 (22) | 67 (23) | <**0.001** |
| **Peridomiciliary** | | | | | | | | |
| Open sewer | 2,81 | 955 (34) | 74 (20) | 131 (39) | 97 (32) | 539 (36) | 114 (39) | <**0.001** |
| Access to the house is paved? | 2,81 | 2,107 (75) | 325 (89) | 259 (78) | 239 (79) | 1,121 (74) | 163 (56) | <**0.001** |
| Wall plastered? | 2,81 | 297 (11) | 19 (5.2) | 40 (12) | 19 (6.2) | 145 (9.6) | 74 (25) | <**0.001** |
| **Presence of animals in the household** | | | | | | | | |
| Cats | 2,81 | 693 (25) | 82 (22) | 79 (24) | 75 (25) | 390 (26) | 67 (23) | 0.60 |
| Dogs | 2,81 | 1,092 (39) | 154 (42) | 108 (33) | 140 (46) | 544 (36) | 146 (50) | <**0.001** |
| Chickens | 2,81 | 138 (4.9) | 29 (7.9) | 14 (4.2) | 36 (12) | 41 (2.7) | 18 (6.1) | <**0.001** |
| *Leptospira* seropositivity | 2,81 | | | | | | | 0.97 |
| Cynopteri | | 45 (1.6) | 6 (1.6) | 7 (2.1) | 3 (1.0) | 23 (1.5) | 6 (2.0) | |
| Icterohaemorrhagiae | | 208 (7.4) | 24 (6.5) | 24 (7.2) | 23 (7.6) | 114 (7.5) | 23 (7.8) | |
| Without evidence of seropositivity | | 2,555 (91) | 337 (92) | 301 (91) | 278 (91) | 1,375 (91) | 264 (90) | |

[1]Mean (SD) or Frequency (%)

[2]Pearson's Chi-squared test

years (OR 1.02 per year, 95% CI 1.01–1.03), work-related exposures (OR 1.71, 95% CI 1.10–2.66) and contact with sewage (OR 1.42, 95% CI 1.00–2.03) were factors associated with seropositivity against Icterohaemorrhagiae.

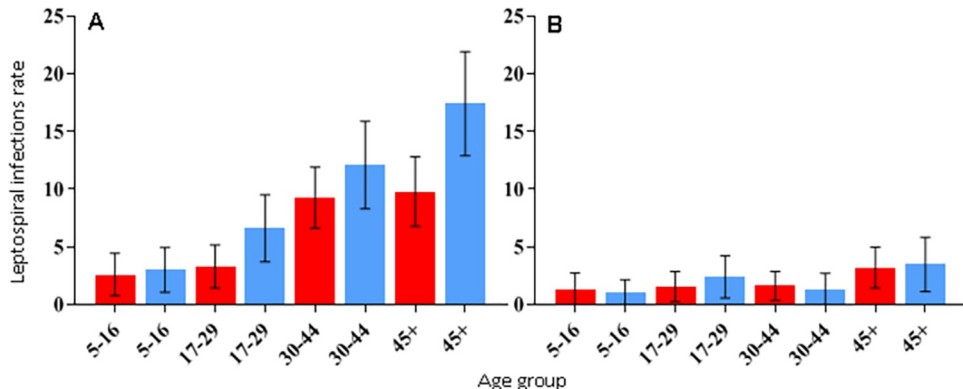

**Fig 2.** Infection rates for *Leptospira interrogans* Icterohaemorrhagiae (A) and *Leptospira kirschneri* Cynopteri (B) by sex and age. Red bars: female. Blue bars: males. Whiskers: 95% CI.

Fig 3 shows differences in the spatial patterns of seropositivity between serogroups Icterohaemorrhagiae and Cynopteri, where yellow rectangle indicate non-overlapping kernel ratio concentration areas between serogroups. The objective of this complementary analysis was to verify possible overlaps in the spatial distributions of both the two serogroups.

## Discussion

In this article, our MAT results indicated that *L. interrogans* serogroup Icterohaemorrhagiae is the most predominant in our areas of study in the city of Salvador, Brazil. In the local context, based on MAT, we found few circulating serogroups, and there was little evidence of cross-reaction (S1 and S2 Tables). Taken together, those results are in accordance with previous serosurveillance and ecological studies conducted in the same region [18,22,23], and supported by the high number of local clinical isolates obtained from humans and rats, where the

**Table 2. Multivariate multinomial regression of seropositivity for *Leptospira kirschneri* Cynopteri (3522C) and *Leptospira interrogans* Icterohaemorrhagiae (Fiocruz L1-130), (n = 2,009).** Note that there were no differences in seropositivity among neighborhoods.

| Characteristics | Cynopteri (3522C) | | Icterohaemorrhagiae (Fiocruz L1-130) | |
|---|---|---|---|---|
| | OR 95% CI | | | |
| Gender (male) | - | - | 1.51 | 1.09–2.10 |
| Age (years) | 1.03 | 1.01–1.06 | 1.02 | 1.01–1.03 |
| Work-related exposures | - | - | 1.71 | 1.10–2.66 |
| Contact with sewage | - | - | 1.42 | 1.00–2.03 |
| Lives in house with unplastered walls (exposed brick) | 1.68 | 1.09–2.59 | - | - |
| **Neighborhood** | | | | |
| Alto do Cabrito (reference) | | | | |
| Marechal Rondon | 0.79 | 0.42–1.47 | 1.54 | 0.43–5.46 |
| Nova Constituinte | 0.98 | 0.51–1.88 | 0.97 | 0.21–4.44 |
| Pau da Lima | 1.19 | 0.74–1.90 | 1.39 | 0.46–4.15 |
| Rio Sena | 1.30 | 0.69–2.45 | 2.46 | 0.67–9.10 |
| Cats near the household | 2.00 | 1.03–3.88 | - | - |

Acronyms: OR, odds ratio; 95% IC, Confidence intervals.

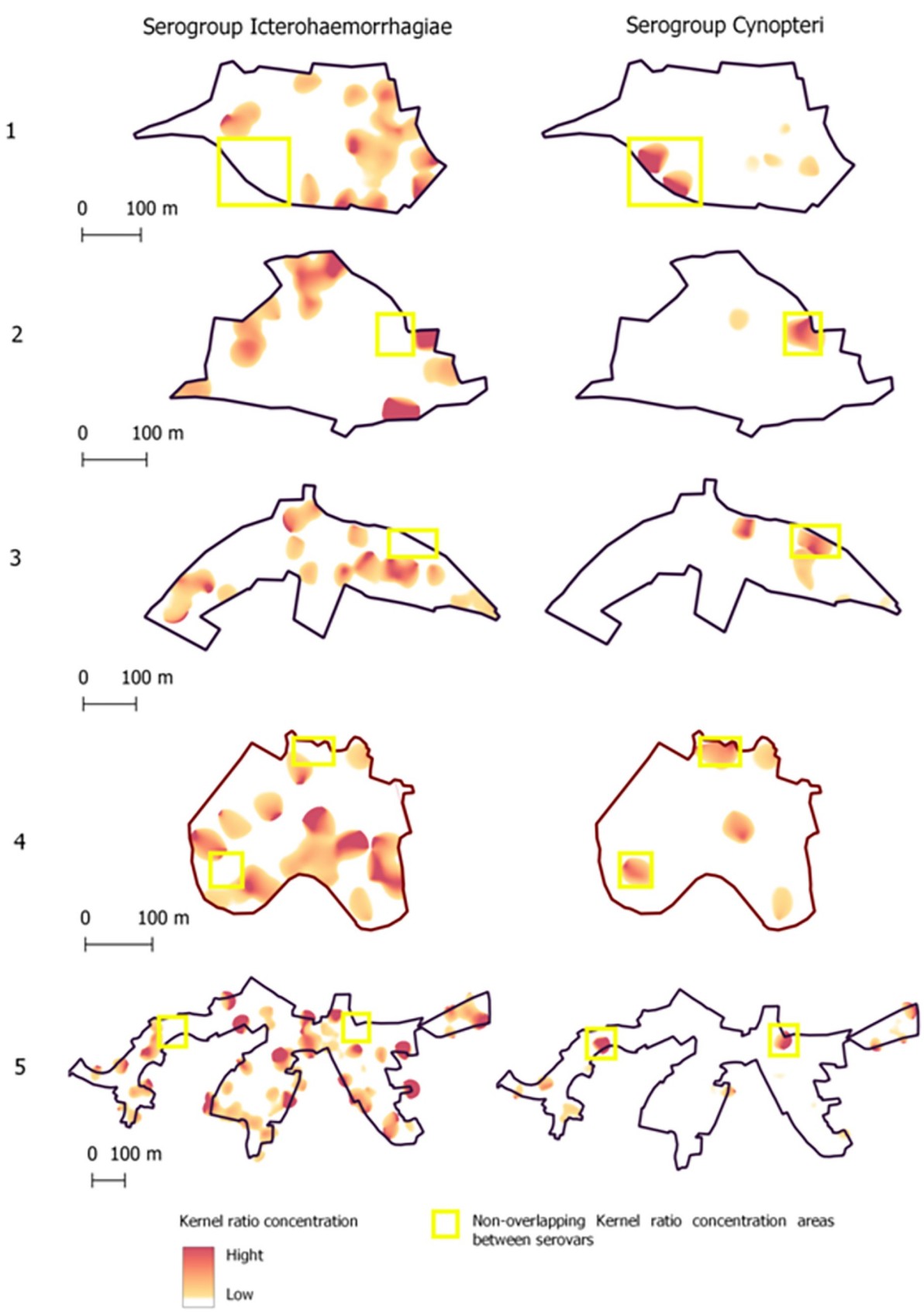

**Fig 3. Spatial patterns of seropositivity to serogroups Icterohaemorrhagiae and Cynopteri (yellow rectangle) on five neighborhoods of Salvador, Bahia, Brazil.** The light red-to- hard red gradient represents increasing density in smoothing analyses which used 30 meters as the bandwidth.

majority of those were identified as serogroup Icterohaemorrhagiae [14,24]. However, we also found evidence of local exposure to *L. kirschneri* serogroup Cynopteri among residents of all the communities studied. Although MAT is a technique with limitations to precisely identify the infecting serogroup, the identification of individuals with high titers against the Cynopteri serogroup, considered pathogenic and whose reservoir in urban environments is unknown, represents an important epidemiological opportunity in attempting to understand the infection patterns of these two important serogroups of pathogenic *Leptospira* that seems to be circulating in this region.

We identified differences in the demographic and socioenvironmental factors that affect seropositivity for each serogroup. Residents exposed to serogroup Cynopteri had a different demographic profile from that of serogroup Icterohaemorrhagiae. We observed higher seroprevalence for serogroup Cynopteri in ages >45 years for both genders, suggesting that behavioral differences between men and women do not affect exposure, in contrast to what has been observed regarding overall seropositivity in previous studies. Men tend to be more engaged in outdoor activities that are considered environmental risk exposures (e.g.: sewer canal cleaning, construction work, debris and refuse management) that are known risk factors for seroprevalence [18,23]. However, for serogroup Icterohaemorrhagiae, we found that seropositivity increased with age, and that males were more likely to be seropositive. A similar relationship between age and exposure to Icterohaemorrhagiae has been observed in previous studies [18].

Occupation-related factors were associated with exposure to serogroup Icterohaemorrhagiae. Individuals who engaged in activities involving solid waste management, or contact with mud, floodwaters and/or sewage were more likely to be seropositive. This agrees with previous results from these and similar communities, where leptospirosis is associated with work-related exposures, including occupation in subsistence farming [3]. Further, inhabitants of low-income communities are commonly enrolled in informal employment, often close to their households. This finding highlights the role of the environment as an important source for transmission. We did not detect any independent associations between work-related risk exposures and seropositivity for Cynopteri. This negative finding may be a consequence of the lower number of individuals seropositive for Cynopteri.

Little is known about the animal and environmental reservoirs for serogroup Cynopteri, other than that the strain was first isolated from a bat kidney in Indonesia [34]. In our final model, we detected a positive association between anti-Cynopteri antibodies and the presence of cats in the household. Although occurrence of this serogroup has been reported in wild animals such as bats [35], there are also indications that cats can be exposed to *Leptospira*; in particular, higher seroprevalence of serogroup Cynopteri than other serogroups has been found in cats from Spain [36]. This suggests a need for future studies to accurately identify the different pathogenic serogroups that circulate in low-income areas and to evaluate the maintenance animal host of the Cynopteri serogroup in these urban environments.

Among the domestic and peri-domiciliary variables, the presence of open sewage near the home was associated with greater seropositivity for serogroup Icterohaemorrhagiae. This result is consistent with a previous study [22] in which "living near an open sewer" was a factor associated with seropositivity and suggests that environmental exposure is an important route of indirect transmission. It should also be noted that the urine of animals other than rats can release leptospires into the environment. This association has also been reported in other

epidemiological studies [37,38]. A previous study in the same study area identified that, in addition to sewage, accumulations of rainwater on the ground can be a source of pathogenic *Leptospira* [39].

Infrastructure deficiencies in homes, for example living in a house with unplastered walls (exposed brick), have also been considered a source of transmission for repeated exposure to *Leptospira* [22]. Risk factors for peri-domiciliary exposure may therefore also be important for infection by serogroup Cynopteri. Structural precariousness in the household could be an indicator of low socioeconomic status, as suggested in another study [25].

The differences that we found in the spatial distribution of Icterohaemorrhagiae and Cynopteri (Fig 3) suggest that these two r serogroups occupy distinct niches within the urban environment. For example, our statistical analysis found that serogroup Icterohaemorrhagiae is associated with proximity to sewers and Cynopteri with the presence of cats.

There are limitations to this work that must be acknowledged. Firstly, whilst MAT is the standard test used in prevalence surveys and can, to some extent, help to indicate the presumptive serogroups circulating in a specific area [27] A positive result does not indicate that the individual necessarily has an active infection unless paired samples are taken [13]. Furthermore, as we collected data at a single point in time we cannot assess-incidence of infection events over time. Another important aspect to be considered is that, although we had previously tested a broader set of seven serogroups, we recognized the possible existence of other serogroups of *Leptospira* in the study population. However, our main focus was not on a comprehensive assessment of all serogroups present in each region, but on identifying the factors that could be associated with the seropositivity of the most highly prevalent serogroups of *Leptospira* in the populations under study. Additionally, we found a low prevalence of Cynopteri, making it difficult to understand its true epidemiological pattern. Future studies may be needed to deepen our understanding of the joint spatial distributions of multiple serogroups in relation to the abundance and mobility of rodents or felines, and proximity to households.

## Conclusion

Our data suggest distinct epidemiological and spatial patterns associated with the Icterohaemorrhagiae and Cynopteri serogroups in the urban environment at high risk for leptospirosis. We emphasize the need for studies that accurately identify the different pathogenic serogroups that circulate and infect residents of low-income areas. Future studies need to further evaluate the role of cats in *Leptospira* transmission and identify the main maintenance animal host of the Cynopteri serogroup in urban environments.

## Supporting information

**S1 Table. Leptospira seropositivity by serogroup.**
(XLSX)

**S2 Table. Stratification Leptospira seropositivity.**
(XLSX)

**S3 Table. Bivariate models for seroprevalence of Leptospira kirschneri Cynopteri (3522C) and Leptospira interrogans Icterohaemorrhagiae (Fiocruz L1-130).**
(DOCX)

**S1 Database. Excel database containing the data used in the tables and figures.**
(XLSX)

## Acknowledgments

The authors would like to thank the individuals from the communities who participated in this study.

## Author Contributions

**Conceptualization:** Daiana de Oliveira, Hussein Khalil, Federico Costa.

**Data curation:** Hussein Khalil, Nivison Nery, Jr, Gielson Almeida do Sacramento, Peter J. Diggle, Federico Costa.

**Formal analysis:** Daiana de Oliveira, Hussein Khalil, Nivison Nery, Jr, Juan C. Quintero-Vélez, Caio Graco Zeppelini, Ricardo Lustosa, Igor Santana Ferreira, Peter J. Diggle, Elsio A. Wunder, Jr, Federico Costa.

**Investigation:** Daiana de Oliveira, Fabiana Almerinda G. Palma, Gielson Almeida do Sacramento, Jaqueline S. Cruz, Ricardo Lustosa, Ticiana Carvalho-Pereira, Peter J. Diggle, Elsio A. Wunder, Jr, Albert I. Ko, Yeimi Alzate Lopez, Mike Begon, Mitermayer G. Reis, Federico Costa.

**Methodology:** Daiana de Oliveira, Hussein Khalil, Fabiana Almerinda G. Palma, Roberta Santana, Nivison Nery, Jr, Jaqueline S. Cruz, Ricardo Lustosa, Igor Santana Ferreira, Ticiana Carvalho-Pereira, Peter J. Diggle, Elsio A. Wunder, Jr, Albert I. Ko, Yeimi Alzate Lopez, Mike Begon, Mitermayer G. Reis, Federico Costa.

**Software:** Nivison Nery, Jr.

**Supervision:** Hussein Khalil, Albert I. Ko, Mike Begon, Mitermayer G. Reis, Federico Costa.

**Visualization:** Caio Graco Zeppelini.

**Writing – original draft:** Daiana de Oliveira, Fabiana Almerinda G. Palma, Nivison Nery, Jr, Caio Graco Zeppelini, Elsio A. Wunder, Jr, Albert I. Ko, Mike Begon, Federico Costa.

**Writing – review & editing:** Daiana de Oliveira, Hussein Khalil, Fabiana Almerinda G. Palma, Nivison Nery, Jr, Juan C. Quintero-Vélez, Jaqueline S. Cruz, Ricardo Lustosa, Peter J. Diggle, Elsio A. Wunder, Jr, Albert I. Ko, Yeimi Alzate Lopez, Mike Begon, Mitermayer G. Reis, Federico Costa.

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
