## [Decision Letter · Decision Letter 0]

24 Jul 2023

Dear Prof Costa,

Thank you very much for submitting your manuscript "Factors associated with differential seropositivity to Leptospira interrogans and Leptospira kirschneri in a high transmission urban setting for leptospirosis in Brazil" for consideration at PLOS Neglected Tropical Diseases. As with all papers reviewed by the journal, your manuscript was reviewed by members of the editorial board and by several independent reviewers. In light of the reviews (below this email), we would like to invite the resubmission of a significantly-revised version that takes into account the reviewers' comments. 

In addition to addressing the reviewers' questions, in particular regarding methodological and results details in study design and analysis, please address the following: 

MAT is the reference serological test but not a gold standard (and not “golden”) which implies perfect sensitivity and specificity. Please change to reference test. 

Regarding the use of MAT as the primary method for the study. In Line 90, authors indicate that MAT can indicate presumptive infective serogroup. We generally use that statement as a word of “caution”, to convey that it is a far we should go in terms of attempting to interpret MAT results since MAT cannot identify serogroups and certainly not serovars. The short sentence in the introduction seems to imply that MAT is a tool validated for identifying serogroups and I believe this needs to be justified further in the context of this specific study and application. Presumptively means that there is a lot of room for misclassification which will affect results. 

The use of MAT in the specific study in Salvador is fairly unique because of the long-term research and understanding of the local ecology indicating very few circulating strains and serogroups which would facilitate interpretation of MAT results. This would not be applicable to other locations with many circulating strains. Authors do explain the study is examining serologic profiles which is a valuable tool but I recommend expanding that introduction to clearly explain that MAT does not identify serogroups, that presumptive serogroups can be reported under certain assumptions, but it can be useful in some contexts such as population level examination of serologic profiles, in particular to detect temporal or spatial shifts in the profiles. Methods need to clearly specify the assumptions for interpretation of MAT and the discussion should include limitations due to potential MAT misclassifications and missing detection of other strains.

We cannot make any decision about publication until we have seen the revised manuscript and your response to the reviewers' comments. Your revised manuscript is also likely to be sent to reviewers for further evaluation.

Sincerely,

Claudia Munoz-Zanzi

Guest Editor

Joseph Vinetz

Section Editor

In addition to addressing the reviewers' questions, in particular regarding methodological and results details in study design and analysis, please address the following: 

MAT is the reference serological test but not a gold standard (and not “golden”) which implies perfect sensitivity and specificity. Please change to reference test. 

Regarding the use of MAT as the primary method for the study. In Line 90, authors indicate that MAT can indicate presumptive infective serogroup. We generally use that statement as a word of “caution”, to convey that it is a far we should go in terms of attempting to interpret MAT results since MAT cannot identify serogroups and certainly not serovars. The short sentence in the introduction seems to imply that MAT is a tool validated for identifying serogroups and I believe this needs to be justified further in the context of this specific study and application. Presumptively means that there is a lot of room for misclassification which will affect results. 

The use of MAT in the specific study in Salvador is fairly unique because of the long-term research and understanding of the local ecology indicating very few circulating strains and serogroups which would facilitate interpretation of MAT results. This would not be applicable to other locations with many circulating strains. Authors do explain the study is examining serologic profiles which is a valuable tool but I recommend expanding that introduction to clearly explain that MAT does not identify serogroups, that presumptive serogroups can be reported under certain assumptions, but it can be useful in some contexts such as population level examination of serologic profiles, in particular to detect temporal or spatial shifts in the profiles. Methods need to clearly specify the assumptions for interpretation of MAT and the discussion should include limitations due to potential MAT misclassifications and missing detection of other strains.

Reviewer's Responses to Questions

**Key Review Criteria Required for Acceptance?**

**Methods**

-Are the objectives of the study clearly articulated with a clear testable hypothesis stated?

-Is the study design appropriate to address the stated objectives?

-Is the population clearly described and appropriate for the hypothesis being tested?

-Is the sample size sufficient to ensure adequate power to address the hypothesis being tested?

-Were correct statistical analysis used to support conclusions?

-Are there concerns about ethical or regulatory requirements being met?

Reviewer #1: Overall, the study demonstrates a clear articulation of objectives, employs an appropriate study design, and describes a suitable population. The study recruited a large sample size and employed robust statistical analyses, which adequately support the claims.

Reviewer #2: Why was Pau da Lima surveyed so much later than the other neighborhoods? I do have some concerns about this given that it also makes up roughly half of the total sample size. Table 1 mostly alleviates these concerns, but this should be discussed especially given what occurred between 2018 and 2020, namely COVID-19. Are we sure that these are comparable populations.

Line 149: How was eligibility determined?

Line 149: What is meant by “free consent”? Is this informed consent? Also, were eligible individuals offered any incentive to participate?

Line 149: How were eligible individuals identified and approached, e.g. door-to-door visits?

Line 151: Please include a sample questionnaire as supplemental information or a citation if it was used previously.

Please include both the precise starting statistical model and the model after variable selection for both analyses. It is unclear to me which variables were initially included and which were excluded by the stepwise variable selection.

I have concerns about the initial screen of variables using bivariate analysis and a p<0.15 threshold. Variables that are not statistically significant in bivariate analysis can be in multivariate analysis. It would be better to include all biologically relevant variables and allow the stepwise variable selection to choose, assuming the model will converge with all variables.

The authors mention using a mixed-effect model, but don’t mention a random effect.

Please include the standard diagnostic plots for all statistical models as supplemental information so that the appropriateness of the models can be determined.

**Results**

-Does the analysis presented match the analysis plan?

-Are the results clearly and completely presented?

-Are the figures (Tables, Images) of sufficient quality for clarity?

Reviewer #1: The analysis presented in the manuscript matches the analysis plan described in the methods section. The results are clearly and completely presented in the manuscript. 

In Table 1, the study presents three outcomes for Leptospira seropositivity: "Icterohaemorrhagiae," "Cynopteri," and "No infection." However, the term "No infection" may not be the most appropriate label in this context. It implies the absence of infection by any serogroups, but the study only used two strains in the MAT to determine seropositivity. Therefore, there is a possibility that the samples could have been infected with serogroups not included in the MAT. It would be helpful if the author could clarify this point or acknowledge it as a potential limitation in the study.

Reviewer #2: In Table 1: Would “Wall material?” be better described as “Wall Plastered?”

Figure 1: This figure or the data used to create needs a citation.

Line 237: Where any individuals seropositive for both?

Line 243: Did any of the neighborhoods have significantly different seropositivity rates from the others before adjusting for other covariates?

Line 245: You state that seroprevalence of Icterohaemorrhagiae increased with age group, what precisely do you mean here? And what is the p-value associated with the associated test of significance?

Figure 2:

 The axes need labels as do panel A and B.

 Are these seropositivity rates as a percentage of total population?

 The upper bound of the male 45+ group on the left plot was cut off

 Please add labels showing which groups are significantly different from others

 Please change gender to sex

Line 266: What model was used for each analysis? Did the analysis of age and sex shown previously include only those variables or other covariates? What variables were included in this analysis?

Line 272: What age group was the OR given for and what was the comparator group?

Table 2: What do the dashes denote? Are they variables that were not included in the final model?

**Conclusions**

-Are the conclusions supported by the data presented?

-Are the limitations of analysis clearly described?

-Do the authors discuss how these data can be helpful to advance our understanding of the topic under study?

-Is public health relevance addressed?

Reviewer #1: Overall, the authors have presented a comprehensive analysis and discussion of the data, ensuring the conclusions are well-supported. They provide a clear understanding of how these findings can serve as a foundation for future research, emphasizing the significant public health implications of the study. 

It would be helpful to expand and elaborate further on the limitation section in the discussion. This could involve addressing any potential biases or confounding factors that may have influenced the results. Additionally, limitations related to the study design, data collection methods, and generalizability of the findings should be acknowledged.

Reviewer #2: Line 305: If feel like the statement “suggesting that behavioral differences between men and women do not affect exposure” is too strong of a statement given that behavioral differences in men and women were not explored in the analysis. If there are specific covariates that are expected to vary between men and women, these should be discussed and evaluated.

Line 360: The authors only explored the spatial niches of seroprevalence, not of the reservoirs.

**Editorial and Data Presentation Modifications?**

Reviewer #1: None.

Reviewer #2: Line 102: I’m not sure what you mean by an “informal accelerated expansion”.

Line 190: possible typo – should this read “as complementary analyses to evaluate”.

Line 191: I’m not sure what the last part of the sentence after “with” is referring to.

Line 319: possible typo – should “maybe because have a” be “maybe due to a”?

Line 354: suggested rewording – “which might not be possible to understand the true” might be better written as “might make it difficult to understand the true”

**Summary and General Comments**

Reviewer #1: The main claims of the paper are to identify the factors associated with seropositivity to Leptospira serogroup Icterohaemorrhagiae and serogroup Cynopteri in a high transmission urban setting for leptospirosis in city of Salvador, Brazil, and to analyze the spatial distribution patterns of these serogroups within the study area. These claims are novel and significant as they contribute to understanding the epidemiology of leptospirosis, specifically the risk factors and spatial distribution of different pathogenic serogroups in urban environments. While previous studies have investigated leptospirosis in similar contexts, this study's specific focus on these two serogroups and their associations with demographic, socioeconomic, and environmental factors adds novelty to the field.

Reviewer #2: I thank the author’s for their submission “Factors associated with differential seropositivity to Leptospira interrogans and Leptospira kirschneri in a high transmission urban setting for leptospirosis in Brazil”. Overall, I found the study well designed, the analysis appropriate, and the paper well written. I do believe that the paper could benefit from several revisions. Most importantly, I believe the methods would benefit from significantly more details about the models used as it not currently possible to replicate their results.

PLOS authors have the option to publish the peer review history of their article (what does this mean?). If published, this will include your full peer review and any attached files.

Reviewer #1: No

Reviewer #2: No

Figure Files:

While revising your submission, please upload your figure files to the Preflight Analysis and Conversion Engine (PACE) digital diagnostic tool, https://pacev2.apexcova

---

## [Decision Letter · Decision Letter 1]

19 Jan 2024

Dear Prof Costa,

Thank you very much for submitting your manuscript "Fatores associados à soropositividade diferencial para Leptospira interrogans e  Leptospira kirschneri em ambiente urbano de alta transmissão para leptospirose no Brasil." for consideration at PLOS Neglected Tropical Diseases. As with all papers reviewed by the journal, your manuscript was reviewed by members of the editorial board and by several independent reviewers. The reviewers appreciated the attention to an important topic. Based on the reviews, we are likely to accept this manuscript for publication, providing that you modify the manuscript according to the review recommendations. 

Reviewer 3:

Please edit the submission form to eliminate non-English.

The manuscript continues to need English language corrections. For example, "unplaster" should be "unflustered." For example, "Our data suggests" should be changed to "our data suggest."Please go through the entire manuscript again to correct spelling, word choice/diction and grammar errors. This journal does not have copyeditors to do this task. 

More importantly, this manuscript continues to appear to conflate serogroup seropositivity with actual infecting serovar. For example, in the abstract, "Our data suggests distinct epidemiological 73 patterns associated with serogroups

74 Icterohaemorrhagiae and Cynopteri within the high-risk urban environment for 75 leptospirosis and with differences of spatial niches." "Future studies must identify the different pathogenic serogroups circulating in low-income areas, and further evaluate the potential role of cats in the transmission of the serogroup Cynopteri in urban settings." What the field really needs is precise identification of the actual infecting Leptospira, either by obtaining an isolate for characterization or sequencing-based identification. Cats involved in leptospirosis transmission? Is this statement an error? It is presented as a conclusion. Perhaps "cats" are a confounder for another variable such as another transmitting species such as rodents. Cats are not thought to be important in leptospirosis ecology; no specific data are presented to support this assertion. As the authors state, "Little is known about the animal and environmental reservoirs for serogroup Cynopteri." This is true, but statistical associations should be treated in a much more circumspect way.

Generally speaking the manuscript can draw statistical conclusions based on seropositivity diagnosing leptospiral exposure but must refrain from drawing causal inference about the infecting strain, unless data are presented to confirm infecting strain.

The manuscript neither cites not discusses the following seminal manuscripts:

Levett PN. Leptospirosis. Clin Microbiol Rev. 2001 Apr;14(2):296-326. doi: 10.1128/CMR.14.2.296-326.2001. PMID: 11292640; PMCID: PMC88975.

Levett PN. Usefulness of serologic analysis as a predictor of the infecting serovar in patients with severe leptospirosis. Clin Infect Dis. 2003 Feb 15;36(4):447-52. doi: 10.1086/346208. Epub 2003 Jan 29. PMID: 12567302.

Abstract

The diagnosis of leptospirosis is often made using the microscopic agglutination test (MAT), in which live antigens representing >20 serogroups undergo reaction with patient serum samples to detect agglutinating antibodies. Data derived from this assay are often used to infer the identity of the infecting leptospiral serovar or serogroup; however, paradoxical reactions and cross-reactions between serogroups are common. To evaluate the usefulness of this approach, data on culture-proven cases of leptospirosis that occurred in Barbados from January 1980 through December 1998 were reviewed. A total of 151 isolates of 4 serovars were identified. The sensitivity of MAT for the prediction of the infecting serovar was determined. Overall, the predominant serogroup at a titer of >or=100 correctly predicted 46.4% of all serovars isolated. If a titer of >or=800 was used as the cutoff, sensitivity decreased slightly to 44.4%. The overall specificity for all serogroups was 64.8%. Serologic analysis appeared to be of little value for the identification of the infecting serovar in individual cases of leptospirosis in humans. Presumptive serogroup reactivity data should be used only to gain a broad idea of the serogroups present at the population level.

Please add a paragraph dedicated to delineating and discussing the limitations of the present work. It should start out something along the lines of "There are limitations to this work that should be considered in the context of the study design and data obtained....."

Information provided regarding serogroup Cynopteri is insufficient for the reader, given the important focus on this serogroup. Most publications citing Cynopteri cite L. interrogans server Cynopteri. What is the basis of L. kirschneri Cynopteri. A comprehensive but concise review of the original isolate of Cynopteri is important as well as information about the use of this serovar/serogroup in the literature.

Sincerely,

Joseph M. Vinetz

Section Editor

Joseph Vinetz

Section Editor

Reviewer 3:

Please edit the submission form to eliminate non-English.

The manuscript continues to need English language corrections. For example, "unplaster" should be "unflustered." For example, "Our data suggests" should be changed to "our data suggest."Please go through the entire manuscript again to correct spelling, word choice/diction and grammar errors. This journal does not have copyeditors to do this task. 

More importantly, this manuscript continues to appear to conflate serogroup seropositivity with actual infecting serovar. For example, in the abstract, "Our data suggests distinct epidemiological 73 patterns associated with serogroups

74 Icterohaemorrhagiae and Cynopteri within the high-risk urban environment for 75 leptospirosis and with differences of spatial niches." "Future studies must identify the different pathogenic serogroups circulating in low-income areas, and further evaluate the potential role of cats in the transmission of the serogroup Cynopteri in urban settings." What the field really needs is precise identification of the actual infecting Leptospira, either by obtaining an isolate for characterization or sequencing-based identification. Cats involved in leptospirosis transmission? Is this statement an error? It is presented as a conclusion. Perhaps "cats" are a confounder for another variable such as another transmitting species such as rodents. Cats are not thought to be important in leptospirosis ecology; no specific data are presented to support this assertion. As the authors state, "Little is known about the animal and environmental reservoirs for serogroup Cynopteri." This is true, but statistical associations should be treated in a much more circumspect way.

Generally speaking the manuscript can draw statistical conclusions based on seropositivity diagnosing leptospiral exposure but must refrain from drawing causal inference about the infecting strain, unless data are presented to confirm infecting strain.

The manuscript neither cites not discusses the following seminal manuscripts:

Levett PN. Leptospirosis. Clin Microbiol Rev. 2001 Apr;14(2):296-326. doi: 10.1128/CMR.14.2.296-326.2001. PMID: 11292640; PMCID: PMC88975.

Levett PN. Usefulness of serologic analysis as a predictor of the infecting serovar in patients with severe leptospirosis. Clin Infect Dis. 2003 Feb 15;36(4):447-52. doi: 10.1086/346208. Epub 2003 Jan 29. PMID: 12567302.

Abstract

The diagnosis of leptospirosis is often made using the microscopic agglutination test (MAT), in which live antigens representing >20 serogroups undergo reaction with patient serum samples to detect agglutinating antibodies. Data derived from this assay are often used to infer the identity of the infecting leptospiral serovar or serogroup; however, paradoxical reactions and cross-reactions between serogroups are common. To evaluate the usefulness of this approach, data on culture-proven cases of leptospirosis that occurred in Barbados from January 1980 through December 1998 were reviewed. A total of 151 isolates of 4 serovars were identified. The sensitivity of MAT for the prediction of the infecting serovar was determined. Overall, the predominant serogroup at a titer of >or=100 correctly predicted 46.4% of all serovars isolated. If a titer of >or=800 was used as the cutoff, sensitivity decreased slightly to 44.4%. The overall specificity for all serogroups was 64.8%. Serologic analysis appeared to be of little value for the identification of the infecting serovar in individual cases of leptospirosis in humans. Presumptive serogroup reactivity data should be used only to gain a broad idea of the serogroups present at the population level.

Please add a paragraph dedicated to delineating and discussing the limitations of the present work. It should start out something along the lines of "There are limitations to this work that should be considered in the context of the study design and data obtained....."

Information provided regarding serogroup Cynopteri is insufficient for the reader, given the important focus on this serogroup. Most publications citing Cynopteri cite L. interrogans server Cynopteri. What is the basis of L. kirschneri Cynopteri. A comprehensive but concise review of the original isolate of Cynopteri is important as well as information about the use of this serovar/serogroup in the literature.

Reviewer's Responses to Questions

**Key Review Criteria Required for Acceptance?**

**Methods**

-Are the objectives of the study clearly articulated with a clear testable hypothesis stated?

-Is the study design appropriate to address the stated objectives?

-Is the population clearly described and appropriate for the hypothesis being tested?

-Is the sample size sufficient to ensure adequate power to address the hypothesis being tested?

-Were correct statistical analysis used to support conclusions?

-Are there concerns about ethical or regulatory requirements being met?

Reviewer #2: (No Response)

**Results**

-Does the analysis presented match the analysis plan?

-Are the results clearly and completely presented?

-Are the figures (Tables, Images) of sufficient quality for clarity?

Reviewer #2: (No Response)

**Conclusions**

-Are the conclusions supported by the data presented?

-Are the limitations of analysis clearly described?

-Do the authors discuss how these data can be helpful to advance our understanding of the topic under study?

-Is public health relevance addressed?

Reviewer #2: (No Response)

**Editorial and Data Presentation Modifications?**

Reviewer #2: Line 113: extra space between and and the ,

**Summary and General Comments**

Reviewer #2: I thank the authors for their thoughtful feedback and edits. I am happy with the current version of the manuscript and believe it should be accepted for publication.

PLOS authors have the option to publish the peer review history of their article (what does this mean?). If published, this will include your full peer review and any attached files.

Reviewer #2: No

Figure Files:

Data Requirements:

Reproducibility:

References

---

## [Editor Report · Decision Letter 2]

28 Mar 2024

Dear Prof Costa,

Thank you very much for submitting your manuscript "Factors associated with differential seropositivity to Leptospira interrogans and Leptospira kirschneri in a high transmission urban setting for leptospirosis in Brazil." for consideration at PLOS Neglected Tropical Diseases. As with all papers reviewed by the journal, your manuscript was reviewed by members of the editorial board and by several independent reviewers. The reviewers appreciated the attention to an important topic. Based on the reviews, we are likely to accept this manuscript for publication, providing that you modify the manuscript according to the review recommendations. 

It has been increasingly recognized that MAT has limited value for classification. It is widely used and plays an important role in sero-epidemiological studies and serologic surveillance; however, as stated in previous reviews caution is needed when making claims about strain identification and causality. I think the current language in this revised version still emphasizes causality without giving proper context. The local disease ecology may be suitable, better than in other locations, due to may be few circulating strains, but this is not immediately represented in the text. 

I believe that if the definition used and assumptions made are properly described and justified in the Methods section, the rest of the analysis is appropriate. Then, in the discussion section, this limitation can be reiterated and the current statement expanded to discuss the impact on the results. 

Specific changes: 

Line 111: The microscopic agglutination test (MAT) is the reference serological assay for the diagnosis of leptospirosis. This enables identification of the strain involved in the infection. It can also help to indicate the serogroups circulating in a specific area or region, thereby supporting spatiotemporal epidemiological studies and reservoir identification [11]. 

This sentence explicitly states that MAT is used for strain identification, which is not the case. Regarding serogroup identification, “identification of presumptive serogroup” has been used more recently as a way to represent the uncertainty in such classification. 

Line 183: Samples were tested against a panel of seven antigens, including five reference strains (WHO Collaborative Laboratory for Leptospirosis, Royal Tropical Institute, Holland): L. kirschneri serovars Cynopteri strain 3522C and Grippothyphosa strain Duyster; L. interrogans serovars Canicola strain H. Utrecht IV and Autumnalis strain Akiyami A; and L. borUTHgpetersenii serovar Ballum strain MUS 127.

State also the serogroups for those strains since this is what MAT resulting are indicating/approximating. 

Line 196: The outcome of seropositive for leptospirosis was defined as seropositivity against serogroups Cynopteri and/or Icterohaemorrhagiae. Line 200: Seropositivity for a specific serogroup was defined as the one with the highest titer.

Explain here the limitation of using MAT for serogroup classification, that it may only give an idea of the presumptive serogroup, and how MAT results were interpreted to produce the outcome for analysis. For example: highest titer, at least 2 titers higher than the titer for any other serogroup, how it was classified if the highest titer was to more than one serogroup, describe how “negative” (the comparison group) was defined, etc. 

Line 315: In this paper, we found that L. interrogans serogroup Icterohaemorrhagiae was the main serogroup responsible for leptospirosis cases in Salvador. 

This statement needs to be toned down regarding finding that serogroup Icterohaemorrhagiae was the main infecting serogroup. Statement needs to be consistent with the limitation of MAT to identify serogroups. If in the local context of expected few circulating serogroups and the interpretation of results (for example: few individuals had titers to other serogroups, there was little evidence of cross-reactions, etc. – these results are not shown), then authors can elaborate a bit more about the relative importance of the investigated serogroups.

Sincerely,

Claudia Munoz-Zanzi

Guest Editor

Joseph Vinetz

Section Editor

It has been increasingly recognized that MAT has limited value for classification. It is widely used and plays an important role in sero-epidemiological studies and serologic surveillance; however, as stated in previous reviews caution is needed when making claims about strain identification and causality. I think the current language in this revised version still emphasizes causality without giving proper context. The local disease ecology may be suitable, better than in other locations, due to may be few circulating strains, but this is not immediately represented in the text. 

I believe that if the definition used and assumptions made are properly described and justified in the Methods section, the rest of the analysis is appropriate. Then, in the discussion section, this limitation can be reiterated and the current statement expanded to discuss the impact on the results. 

Specific changes: 

Line 111: The microscopic agglutination test (MAT) is the reference serological assay for the diagnosis of leptospirosis. This enables identification of the strain involved in the infection. It can also help to indicate the serogroups circulating in a specific area or region, thereby supporting spatiotemporal epidemiological studies and reservoir identification [11]. 

This sentence explicitly states that MAT is used for strain identification, which is not the case. Regarding serogroup identification, “identification of presumptive serogroup” has been used more recently as a way to represent the uncertainty in such classification. 

Line 183: Samples were tested against a panel of seven antigens, including five reference strains (WHO Collaborative Laboratory for Leptospirosis, Royal Tropical Institute, Holland): L. kirschneri serovars Cynopteri strain 3522C and Grippothyphosa strain Duyster; L. interrogans serovars Canicola strain H. Utrecht IV and Autumnalis strain Akiyami A; and L. borUTHgpetersenii serovar Ballum strain MUS 127.

State also the serogroups for those strains since this is what MAT resulting are indicating/approximating. 

Line 196: The outcome of seropositive for leptospirosis was defined as seropositivity against serogroups Cynopteri and/or Icterohaemorrhagiae. Line 200: Seropositivity for a specific serogroup was defined as the one with the highest titer.

Explain here the limitation of using MAT for serogroup classification, that it may only give an idea of the presumptive serogroup, and how MAT results were interpreted to produce the outcome for analysis. For example: highest titer, at least 2 titers higher than the titer for any other serogroup, how it was classified if the highest titer was to more than one serogroup, describe how “negative” (the comparison group) was defined, etc. 

Line 315: In this paper, we found that L. interrogans serogroup Icterohaemorrhagiae was the main serogroup responsible for leptospirosis cases in Salvador. 

This statement needs to be toned down regarding finding that serogroup Icterohaemorrhagiae was the main infecting serogroup. Statement needs to be consistent with the limitation of MAT to identify serogroups. If in the local context of expected few circulating serogroups and the interpretation of results (for example: few individuals had titers to other serogroups, there was little evidence of cross-reactions, etc. – these results are not shown), then authors can elaborate a bit more about the relative importance of the investigated serogroups.

Figure Files:

Data Requirements:

Reproducibility:

References

---

## [Editor Report · Decision Letter 3]

2 May 2024

Dear Prof Costa,

We are pleased to inform you that your manuscript 'Factors associated with differential seropositivity to Leptospira interrogans and Leptospira kirschneri in a high transmission urban setting for leptospirosis in Brazil.' has been provisionally accepted for publication in PLOS Neglected Tropical Diseases.

Best regards,

Claudia Munoz-Zanzi

Guest Editor

Joseph Vinetz

Section Editor

---

## [Editor Report · Acceptance letter]

10 May 2024

Dear Prof Costa,

We are delighted to inform you that your manuscript, "Factors associated with differential seropositivity to *Leptospira interrogans* and *Leptospira kirschneri* in a high transmission urban setting for leptospirosis in Brazil," has been formally accepted for publication in PLOS Neglected Tropical Diseases.

Best regards,

Shaden Kamhawi

co-Editor-in-Chief

Paul Brindley

co-Editor-in-Chief
